# Electrophoretic Deposition of One- and Two-Layer Compacts of Holmium and Yttrium Oxide Nanopowders for Magneto-Optical Ceramics Fabrication

Elena G. Kalinina [1,2,3], Nataliya D. Kundikova [1,4], Dmitrii K. Kuznetsov [5] and Maxim G. Ivanov [1,3,6,*]

1   Institute of Electrophysics, Ural Branch of Russian Academy of Sciences, 106 Amundsen Str., 620016 Ekaterinburg, Russia; jelen456@yandex.ru (E.G.K.); kundikovand@susu.ru (N.D.K.)
2   Department of Physical and Inorganic Chemistry, Institute of Natural Sciences and Mathematics, Ural Federal University, 19 Mira Avenue, 620002 Ekaterinburg, Russia
3   G.G. Devyatykh Institute of Chemistry of High-Purity Substances, Russian Academy of Sciences, 49 Tropinin Str., 603950 Nizhny Novgorod, Russia
4   Department of Optoinformation, South Ural State University, 76 Lenin Prospekt, 454080 Chelyabinsk, Russia
5   Department of Condensed Matter Physics and Nanoscale Systems, Institute of Natural Sciences and Mathematics, Ural Federal University, 19 Mira Avenue, 620002 Ekaterinburg, Russia; dimak@urfu.ru
6   Center of Excellence for Photoconversion, Vinča Institute of Nuclear Sciences—National Institute of the Republic of Serbia, University of Belgrade, P.O. Box 522, 11001 Belgrade, Serbia
*   Correspondence: max@iep.uran.ru

**Abstract:** In this work, the possibility of fabricating composite magneto-optical ceramics by electrophoretic deposition (EPD) of nanopowders and high-temperature vacuum sintering of the compacts was investigated. Holmium oxide was chosen as a magneto-optical material for the study because of its transparency in the mid-IR range. Nanopowders of magneto-optical $(Ho_{0.95}La_{0.05})_2O_3$ (HoLa) material were made by self-propagating high-temperature synthesis. Nanopowders of $(Y_{0.9}La_{0.1})_2O_3$ (YLa) were made by laser synthesis for an inactive matrix. The process of formation of one- and two-layer compacts by EPD of the nanopowders from alcohol suspensions was studied in detail. Acetylacetone was shown to be a good dispersant to obtain alcohol suspensions of the nanopowders, characterized by high zeta potential values (+29–+80 mV), and to carry out a stable EPD process. One-layer compacts were made from the HoLa and YLa nanopowders with a density of 30–43%. It was found out that the introduction of polyvinyl butyral (PVB) into the suspension leads to a decrease in the mass and thickness of the green bodies deposited, but does not significantly affect their density. The possibility of making two-layer (YLa/HoLa) compacts with a thickness of up to 2.6 mm and a density of up to 46% was demonstrated. Sintering such compacts in a vacuum at a temperature of 1750 °C for 10 h leads to the formation of ceramics with a homogeneous boundary between the YLa/HoLa layers and a thickness of the interdiffused ion layer of about 30 μm.

**Keywords:** magneto-optical ceramics; electrophoretic deposition (EPD); two-layer ceramics; holmium oxide; yttrium oxide

## 1. Introduction

One of the challenges of magneto-optical device fabrication is to increase the maximum allowable power of laser radiation in Faraday isolators [1]. When a laser beam passes through a magneto-optical element, heat is inevitably released and thermally induced polarization and phase distortions appear, which leads to a deterioration in the quality of the laser beam and a decrease in the insulating ability of the device. This problem can be solved by new designs of magneto-optical elements, in particular, by creating a composite structure. Traditionally, magneto-optical elements have a rod configuration with radial cooling. In this case, the higher the power of the transmitted radiation and

the lower the thermal conductivity of the material, the greater the temperature difference between the central and peripheral parts of the optical element, and the greater the thermally induced distortions (thermal lens, etc.). In laser technology, the problem of reducing the thermal lens can be solved by organizing efficient heat removal by creating a composite element consisting of an active medium and an inactive matrix, for example: YAG/Nd:YAG/YAG [2], YAG/Yb:YAG/YAG [3], Yb:YAG/YAG [4,5], Er:YAG/YAG [6], or a material with a noticeably higher thermal conductivity, for example Yb:YAG/sapphire [4] and Nd:YAG/diamond [7]. In these cases, the released heat is effectively removed from the active medium, reducing the radial temperature gradient, which makes it possible to increase the laser radiation power and/or provide less distortion of the laser beam. With the development of optical ceramic technologies, this approach is being used in active laser elements, but we are not aware of the implementation of such an approach for magneto-optical elements.

One of the promising magneto-optical materials to use in the mid-IR range is holmium oxide [1], the possibility of obtaining transparent ceramics of which was recently shown in [8]. Usually, to obtain optically transparent ceramics from sesquioxides, such methods as slip casting [9] and isostatic pressing followed by vacuum sintering [8], hot pressing (HP) [10–12], hot isostatic pressing (HIP) [13–17], and spark plasma sintering (SPS) [18,19] are used. Hot pressing methods require expensive technological equipment and also have a number of disadvantages associated with contamination of the ceramics, appearance of mechanical defects, and internal stresses. Among relatively cheap forming methods that give a chance to obtain the large-size dense unstressed compacts (green bodies) needed for further sintering of the transparent ceramics, one can single out a method of slip [20] or tape [6] casting, based on the preparation of concentrated suspensions of nanoparticles with the help of dispersants and polymer binders. However, it should be noted that the burnout of the polymers at the sintering stage can lead to the formation of light-scattering defects inside the ceramics.

Relatively recently, it was shown that a promising method for the formation of transparent ceramics is the method of electrophoretic deposition (EPD) [21], which has a high deposition rate, the possibility of using powders of various compositions and size distribution, and gives a chance for thickness control (from thin films to bulk compacts) and sample morphology by changing the deposition parameters (applied voltage and current, deposition time). The preparation of stable powder suspensions is an important step in the implementation of the EPD process. The degree of stability of suspensions is characterized by the electrokinetic zeta potential and the value of the zeta potential can be controlled by using dispersants, for example, acetylacetone, polyethyleneimine, ammonium polyacrylate, etc. [22]. Talking about the stabilization of the nanoparticles in the suspensions, it should be noted that when using EPD to obtain dense ceramic green bodies, one has to solve a contradictory task. On the one hand, suspensions must be stable during the entire EPD process, i.e., the particles should repel each other and not flocculate, and on the other hand, after deposition on the electrode, the particles should form an "ideal" densely packed structure. In the works [23,24], a possibility of sintering optical ceramics based on $(La_xY_{1-x})_2O_3$ solid solutions, where lanthanum oxide was used as a sintering aid, and green bodies with a relatively low density (30–40%) made by EPD from isopropyl alcohol suspensions of nanoparticles produced by laser synthesis was shown.

In this work, we systematically studied the processes of the formation of bulk single-layer and two-layer compacts by the EPD method from non-aqueous suspensions using a dispersant (acetylacetone) and a polymeric binder (polyvinyl butyral) based on $(Y_{0.9}La_{0.1})_2O_3$ (YLa) nanopowders obtained by laser synthesis and $(Ho_{0.95}La_{0.05})_2O_3$ (HoLa) obtained by self-propagating high-temperature synthesis (SHS). The main objectives of the research were to determine the effect of the dispersant and the binder on the zeta potential, the current during the EPD, the mass and density of the green bodies, and also to study the possibility of sintering the two-layer ceramics.

## 2. Materials and Methods

### 2.1. Nanopowders and Suspensions of $(Y_{0.9}La_{0.1})_2O_3$

Two batches of nanopowders of $(Y_{0.9}La_{0.1})_2O_3$ (*YLa*) solid solutions were obtained by evaporating the material with ytterbium fiber laser radiation at a wavelength of 1.07 μm. The process is described in detail in [23,25]. The material to evaporate was mixed from commercially available powders of high purity yttrium and lanthanum oxides (99.99% Polirit, Moscow, Russia), pressed into cylindrical targets, 60 mm in diameter and 20 mm thick, and sintered in air at a temperature of 1300 °C for 1 h. The targets were evaporated under the pulsed laser radiation with a pulse repetition rate of 2 kHz, a pulse duration of 60 μs, and an average power of 250 W. The evaporation chamber was blown through with air at atmospheric pressure and a flow rate of 40 L/min. The nanopowders carried away by the air stream were collected in a cyclone and an electrostatic filter. The specific surface areas of the *YLa* nanopowders collected in a cyclone and an electrostatic precipitator were 76 $m^2$/g and 80 $m^2$/g, respectively, and the batches of the nanopowders were accordingly named *76YLa* and *80YLa*. The specific surface area of the *YLa* nanopowders collected in a cyclone and an electrostatic precipitator was determined by the BET method (TriStar 3000, Micromeritics, Norcross, GA, USA). All obtained powders were annealed in air at 900 °C for 3 h in a camber furnace (LHT 02/18, Nabertherm, Lilienthal, Germany). The XRD was recorded with a D8 DISCOVER GADDS (Bruker AXS, Karlsruhe, Germany) with Cu ($K_{\alpha1,2}$ λ = 1542 Å) radiation and a carbon monochromator. The XRD data were analyzed using the TOPAS 3 software (Bruker AXS, Karlsruhe, Germany) with Rietveld's algorithm to specify structure parameters.

To prepare the suspensions, isopropanol (special purity grade, JSC "Component-Reaktiv", Moscow, Russia) (iPrOH) was used as a dispersion medium. Suspensions of 80YLa with an initial concentration of 70 g/L were prepared using an ultrasonic bath (UZV-13/150-TN, RELTEC, Yekaterinburg, Russia) for 250 min. Acetylacetone (analytically pure grade, Merck, Darmstadt, Germany) (AcAc) was used as a dispersant. The concentration of the acetylacetone was chosen based on the calculation of the mass of acetylacetone per total surface area of nanoparticles in suspension according to the formula:

$$m_{AcAc} = \mu \cdot S_{sp} \cdot m_{np}, \tag{1}$$

where $\mu$ is the mass of acetylacetone per unit surface of the nanopowder, mg/$m^2$—the value was 1 mg/$m^2$ in these experiments; $S_{sp}$ is the specific surface area of the nanopowder ($m^2$/g); $m_{np}$ is the mass of the nanopowder (g); and $m_{AcAc}$ is the mass of acetylacetone (mg).

The large particles in *80YLa* suspensions were removed by centrifugation with a Hermle Z383 centrifuge at a speed of 2000 rpm for 1 min. The concentration of the suspension after centrifugation was about 60 g/L. To reduce the probability of crack formation at the stage of drying the compacts, a polymer modifier, polyvinyl butyral (PVB), was introduced into the suspension as a binder in an amount of 1 mg/$m^2$.

Suspensions of *76YLa* at a concentration of 70 g/L were prepared in isopropanol and sonicated with an ultrasonic probe (Bandelin SONOPULS HD 3200, Berlin, Germany) at a frequency of 20 kHz and an ultrasonic power of 40 W for 250 min. After the treatment, AcAc (1 mg/$m^2$) was added. To prepare the *76YLa_milling* suspension, *76YLa* powder was used, which was subjected to two-stage grinding in a planetary mill (MPP-1, VIBROTECHNIK, St. Petersburg, Russia) in isopropyl alcohol with ceramic zirconium balls of 5 mm in diameter at the first stage for 5 h and 2 mm in diameter at the second stage for 5 h, at a rotation speed of 320 rpm. The *76YLa_milling* suspension was prepared in the isopropanol with the addition of acetylacetone (1 mg/$m^2$), followed by ultrasonic treatment for 125 min (UZV-13/150-TN ultrasonic bath) and centrifugation at 2000 rpm for 1 min. The concentration of the deaggregated suspension was 58 g/L.

### 2.2. Nanopowders and Suspensions of (Ho$_{0.95}$La$_{0.05}$)$_2$O$_3$

The (Ho$_{0.95}$La$_{0.05}$)$_2$O$_3$ (*HoLa*) nanopowder was obtained by the method of self-propagating high-temperature synthesis (SHS). The process is described in detail in [8]. The materials used for the SHS were holmium oxide (99.99% Polirit, Moscow, Russia), lanthanum oxide (99.99% Polirit, Moscow, Russia), nitric acid (99.9999%, Khimreaktiv, Nizhnii Novgorod, Russia), and glycine NH$_2$CH$_2$COOH (99.9%, Khimreaktiv, Nizhnii Novgorod, Russia). The powders received after SHS were annealed in air at a temperature of 900 °C for 3 h. The *HoLa* powder was ground in a planetary mill (MPP-1) in isopropyl alcohol with ceramic zirconium balls of 5 mm in diameter at a rotation speed of 320 rpm for 5 h. Suspensions with an initial concentration of 70 g/L were sonicated using an ultrasonic bath (UZV-13/150-TN) for 125 min. Acetylacetone (1 mg/m$^2$) was used as a dispersant. Undestroyed large aggregates were removed by centrifugation of the suspensions with a Hermle Z383 centrifuge at a rotation speed of 1500 rpm for 1 min. The concentration of the deaggregated suspension was 62 g/L.

### 2.3. Characterization of the Suspensions and Electrophoretic Deposition (EPD)

Electrokinetic measurements were performed by the electroacoustic method using a DT-300 analyzer (Dispersion Technology Inc., Bedford Hills, NY, USA). All measurements for suspensions were carried out under isothermal conditions in air at 298 K.

#### 2.3.1. Electrophoretic Deposition of One-Layer Compacts

Electrophoretic deposition was performed on a specialized computerized setup providing constant voltage regimes, which was developed and made at the Institute of Electrophysics, Ural Branch of the Russian Academy of Sciences. EPD was performed in a cell with a horizontal arrangement of electrodes. An aluminum foil disk with an area of 113 mm$^2$ was a cathode and a stainless-steel disk served as an anode. The distance between the electrodes was 10 mm. To form bulk one-layer compacts by EPD from nanoparticle suspensions, the following deposition modes were used: for suspensions of *76YLa* (3 samples) and *80YLa* (2 samples) powders, the constant voltage was 20 V, and for the *HoLa* suspensions (2 samples) it was 40 V. The deposition time in all cases was 120–150 min. DC current was measured with a UT71E multimeter (Uni-Trend Technology (China) Limited, Dongguan, China). The accuracy of the measurement was ±0.1%. During EPD, the suspensions were pumped from the bottom of the cell to its upper part. The green bodies deposited were dried on the cathode for 7 days in isopropyl alcohol vapor using a desiccator.

#### 2.3.2. Electrophoretic Deposition of Two-Layer Compacts

To form two-layer compacts, layer-by-layer EPD was performed from suspensions of either *76YLa* or *80YLa* powders at a constant voltage of 20 V for 120 min; then, EPD was performed from the suspension of the *HoLa* powders at a constant voltage of 40 V for 90–150 min. Two samples of two-layer compacts were formed: *76YLa/HoLa* and *80YLa/HoLa*. During EPD, the suspension was pumped from the bottom of the cell to its upper part. The resulting green bodies were dried on the cathode for 7 days in isopropyl alcohol vapor using a desiccator.

### 2.4. Sintering of the Ceramics

Sintering of the ceramics was carried out in an SNVE 1.3.1–20 vacuum furnace (Prizma, Iskitim, Russia) with a tungsten heater at a pressure of about $1 \times 10^{-2}$ Pa, a heating rate of 0.5 °C/min, a temperature of 1750 °C, and a 10 h holding time at the maximum temperature. As our previous research demonstrated, the sintering conditions gave a chance to make ytterbium-doped (Y$_{0.9}$La$_{0.1}$)$_2$O$_3$ laser ceramics [25] and (Ho$_{0.95}$La$_{0.05}$)$_2$O$_3$ magneto-optical ceramics [8] where the lanthana was used as a sintering aid. Disk samples were cut across the deposited layers and polished with a Phoenix Beta Grinder/Polisher (BUEHLER, Leinfelden-Echterdingen, Germany). Investigation of the samples was carried out with the help of an EVO LS 10 scanning electron microscope (Carl Zeiss, Oberkochen, Germany).

Sample surfaces were visualized using backscattered electron (BSE) and secondary electron (SE) imaging at accelerated voltages of 5 and 20 kV. The elemental composition of samples was examined by energy-dispersive X-ray spectroscopy (EDS) with the Inca Energy X-MAX$^N$ 50 detector (Oxford Instruments, Abingdon, UK).

## 3. Results

### 3.1. Characteristics of $(Y_{0.9}La_{0.1})_2O_3$ Nanopowders and Suspensions

The harvested *YLa* powders contained weakly agglomerated spherical nanoparticles with a characteristic size of 15–20 nm. The morphology of nanoparticles and their size distribution are shown in [24]. In addition to the nanoparticles, the powder contained spherical particles with a characteristic size of 100–200 nm, formed as a result of the spattering of melted material in the focal spot of the laser radiation, and shapeless particles with a size of 10–100 µm, formed due to splitting of a crust from the target surface. The amount of such large particles in the powder collected in the electrostatic filter was about 1% wt., and in the powder collected in the cyclone was up to 10% wt. The micron-sized particles were removed during further processing by centrifugation of suspensions. A batch of *76YLa* powders was sieved with a 200 Mesh polymer sieve prior to further processing. The specific surface area of the *YLa* nanopowders after annealing, determined by the BET method, was about 30 m$^2$/g. XRD analyses of the annealed nanopowders revealed the nanoparticles to be single-phase cubic structures of Y$_2$O$_3$ sesquioxides (space group *Ia-3*, a = 10.68(2) Å).

The powder used to prepare the suspensions, the dispersant, the binder, and the value of the ζ-potential of the nanoparticles of the resulting suspensions are listed in Table 1.

**Table 1.** Composition of the suspensions and ζ-potential of the nanoparticles.

| Powders | Dispersion Medium | ζ–Potential, mV |
|---|---|---|
| | iPrOH | +4 |
| *76YLa* | iPrOH + AcAc (1 mg/m$^2$) | +49 |
| | iPrOH + AcAc (1 mg/m$^2$) + PVB (1 mg/m$^2$) | +29 |
| *76YLa_milling* | iPrOH | +11 |
| | iPrOH + AcAc (1 mg/m$^2$) | +49 |
| | iPrOH | +13 |
| *80YLa* | iPrOH + AcAc (1 mg/m$^2$) | +69 |
| | iPrOH + AcAc (1 mg/m$^2$) + PVB (1 mg/m$^2$) | +67 |
| | iPrOH | +16 |
| *HoLa* | iPrOH + AcAc (1 mg/m$^2$) | +86 |
| | iPrOH + AcAc (1 mg/m$^2$) + PVB (1 mg/m$^2$) | +80 |

The suspension was treated with an ultrasonic probe for 5 min (*76YLa_AcAc*). The concentration of the *76YLa_AcAc* suspension after centrifugation (2000 rpm, 1 min) was 64 g/L. The *76YLa_PVB* suspension was prepared by introducing PVB (1 mg/m$^2$) into the *76YLa_AcAc* suspension after electrophoretic precipitation of the *76YLa_1* sample (Table 2).

Additionally, aging of the *76YLa_PVB* and *80YLa* suspensions for 14 days was studied, which is of interest from the point of view of the practical application of EPD technology, which should withstand the use of stable suspensions for a long time. During the study of the aging, the suspension was stored in a dark place at room temperature for 14 days.

### 3.2. Characteristics of $(Ho_{0.95}La_{0.05})_2O_3$ Nanopowders and Suspensions

The specific surface area of the *HoLa* nanopowder after annealing, determined by the BET method, was 18 m$^2$/g. The powder consisted of a mixture of large particles with sizes of

hundreds of nanometers and small nanometer-size particles. The morphology of nanoparticles of $(Ho_{1-x}La_x)_2O_3$ solid solutions made by SHS and their size distribution are given in [8]. XRD analysis of the nanopowders showed a C-type cubic structure of $Ho_2O_3$ sesquioxides (space group *Ia-3*, No. 206, Z = 16). No second phases were found in the powders. The composition of the suspensions prepared and the $\zeta$-potential of the *HoLa* nanoparticles are given in Table 1. The suspensions used, the EPD parameters, and the characteristics of the obtained one-layer and two-layer compacts are shown in Table 2 and Table 3, respectively.

**Table 2.** Suspensions and parameters of the EPD of one-layer compacts.

| Sample | Suspension | Dispersion Medium | Mode of EPD (Voltage, Time); Mass/Thickness/Density of the Green Body |
|---|---|---|---|
| *76YLa_1* | *76YLa_AcAc* | iPrOH/AcAc | 20 V, 150 min; 440 mg/1.6 mm/43% |
| *76YLa_2* | *76YLa_PVB* aged for 14 days | iPrOH/AcAc/PVB | 20 V, 150 min; 175 mg/1 mm/32% |
| *76YLa_3* | *76YLa_milling* | iPrOH/AcAc | 20 V, 150 min; 524 mg/2.5 mm/37% |
| *80YLa_1* | *80YLa* | iPrOH/AcAc/PVB | 20 V, 120 min; 250 mg/1.5 mm/38% |
| *80YLa_2* | *80YLa* aged for 14 days | iPrOH/AcAc/PVB | 20 V, 120 min; 171 mg/1.0 mm/32% |
| *HoLa_1* | *HoLa_AcAc* | iPrOH/AcAc | 40 V, 120 min; 751 mg/1.9 mm/30% |
| *HoLa_2* | *HoLa_PVB* | iPrOH/AcAc/PVB | 40 V, 120 min; 447 mg/1.5 mm/32% |

**Table 3.** Suspensions and parameters of the EPD of two-layer compacts.

| Sample | Suspension | Dispersion Medium | Mode of EPD (Voltage, Time); Mass/Thickness/Fraction of the Layer | Density of the Green Body, % of Theoretical * |
|---|---|---|---|---|
| *76YLa/HoLa_1* | *76YLa_milling* | iPrOH/AcAc | 20 V, 120 min; 570 mg/1.6 mm/0.62 | 46 |
| | *HoLa_PVB* | iPrOH/AcAc/PVB | 40 V, 150 min; 356 mg/1.0 mm/0.38 | |
| *80YLa/HoLa_2* | *80YLa_PVB* | iPrOH/AcAc/PVB | 20 V, 120 min; 266 mg/0.7 mm/0.58 | 46 |
| | *HoLa_PVB* | iPrOH/AcAc/PVB | 40 V, 90 min; 190 mg/0.5 mm/0.42 | |

* Note: The layer fraction for the two-layer compact is defined as the ratio of the thickness of the deposited layer to the total thickness of the two-layer compact.

## 4. Discussion

To obtain the bulk compacts by electrophoretic deposition, a high aggregation stability of the suspensions is required. In the case of electrostatic stabilization, the stability is directly related to the $\zeta$-potential of nanoparticles. From Table 1, it is seen that the initial suspensions in isopropanol are characterized by low values of the $\zeta$-potential (+4...+16 mV). Previous experiments on the EPD of nanoparticles from isopropanol suspensions [24] showed the necessity to use a dispersant, because without it, when the electric field was applied, the flocculation process began in the suspension and the entire nanopowder settled to the bottom of the cuvette rather quickly. When acetylacetone, used in this study as a dispersant, is added, the zeta potential of nanoparticles increases significantly (+49...+86 mV), which

is due to an increase in the surface charge of the particles because of the adsorption of protons formed during the dissociation of the enol form of acetylacetone on the surface of the nanoparticles [26]. When PVB, used as a binder, is added to the suspension, the zeta potential of the nanoparticles decreases (Table 1). The authors of [27] reported a similar effect of reducing the zeta potential when PVB was added to a suspension based on zirconium dioxide. The decrease in the $\zeta$-potential may be due to the adsorption of PVB molecules on the particle surface, which prevents the adsorption of protons. At the same time, despite the effect of PVB on the electrokinetic parameters, the values of the $\zeta$-potential still exceeded, in absolute value, the characteristic value of 26 mV, which is necessary for the stable conduction of the EPD process [28].

### 4.1. Electrophoretic Deposition of Single-Layer Compacts from Suspensions of the YLa Nanopowders

Figure 1 shows dependences of the current strength on the time of EPD from suspensions of *76YLa* powder in isopropanol with acetylacetone (1 mg/m$^2$) without a binder (*76YLa_AcAc*) and with the addition of PVB (*76YLa_PVB*). It can be seen that the dependences of the current strength on the EPD time do not have a clearly pronounced downward trend associated with the depletion of the suspensions during the EPD process, which indicates a predominantly ionic nature of the charge transfer. Moreover, an increase in current (about 10%) was observed in the *76YLa_AcAc* suspension for about 60 min, which is associated with an increase in the concentration of ions in the medium, apparently due to processes occurring at the electrodes. Only after 110 min of EPD did the processes of increasing the electrical resistance of the layer of nanoparticles deposited on the cathode and reducing the concentration of the nanoparticles remaining in the suspension begin to affect the current value. The significant effect of the PVB addition on the conductivity of the suspension (curve 2, Figure 1) is obviously associated both with a decrease in the $\zeta$-potential of nanoparticles (the values of the $\zeta$-potential and electrophoretic mobility are directly proportional, according to the Henry equation [29]) and with higher resistivity of the layer deposited. It can be seen that in the first minutes (sometimes tens of seconds) of the formation of a layer of nanoparticles on the cathode, there is a rapid drop in the value of the current, and then, for about 40 min, additional reduction of the current by another 20%.

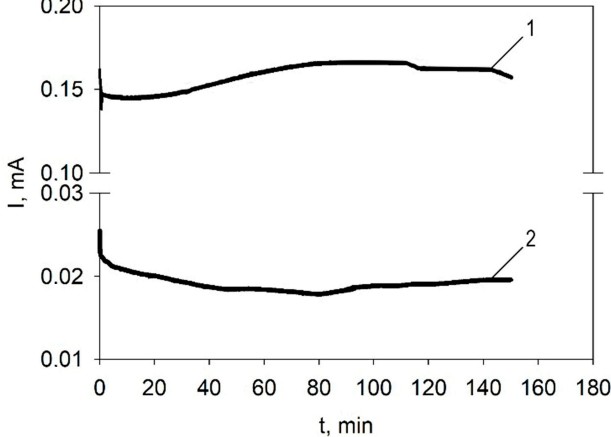

**Figure 1.** Dependence of the current on the EPD time (electric field strength of 20 V/cm) in suspensions: *76YLa_AcAc* without a binder (sample *76YLa_1*, curve 1) and *76YLa_PVB* with PVB addition (sample *76YLa_2*, curve 2).

Figure 2 shows the dependences of the current strength on the time of EPD from suspensions of *80YLa_PVB* in isopropanol with acetylacetone (1 mg/m$^2$) with the addition of PVB binder (1 mg/m$^2$) freshly prepared and aged for 14 days. It can be seen that the dependences have a similar character, for which the current value is approximately constant

for 120 min. It should be noted that when the suspension was aged for 14 days, there was a significant (four-fold) increase in the current (Figure 2, curve 2), which may be due to a change in the ionic composition of the dispersion medium because of the formation of complex compounds, namely metal acetylacetonates with the release of protons, which has a significant effect on the conductivity of the suspension [26]. However, when EPD from *80YLa_PVB* suspension was made, aging of the suspension had a negative effect: there was a decrease in the mass, thickness, and density of the *80YLa_2* sample (Table 2), despite the higher current during EPD (Figure 2).

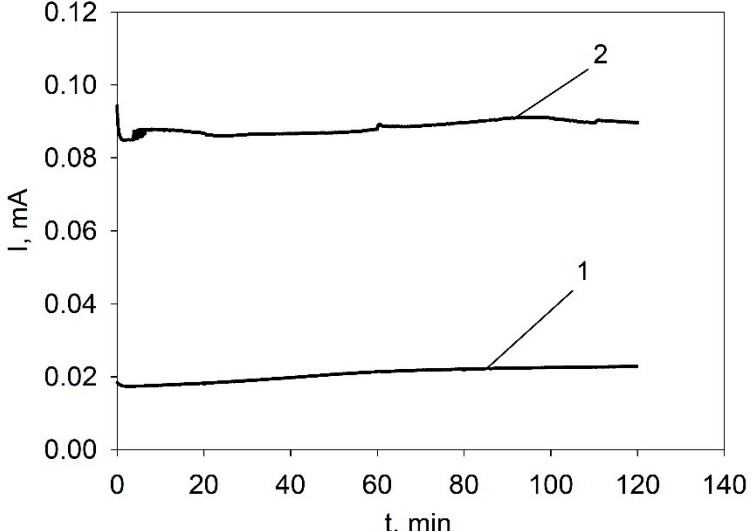

**Figure 2.** Dependences of the current on the time of EPD (electric field strength of 20 V/cm) from *80YLa_PVB* suspension: freshly prepared (sample *80YLa_1*, curve 1) and aged for 14 days (sample *80YLa_2*, curve 2).

Single layer compacts *76YLa_1*, *76YLa_2* and *76YLa_3* were obtained under the same EPD conditions from suspensions of *76YLa_AcAc*, *76YLa_PVB* and *76YLa_milling*, respectively. From Table 2, it can be seen that samples with the largest mass and thickness were obtained from suspensions without the addition of the PVB. The addition of a polymeric binder led to a decrease in the thickness and weight of the samples; however, upon drying, the *76YLa_2* sample deposited from a suspension with the addition of the PVB retained its integrity, in contrast to the samples without the PVB. Sample *76YLa_3*, deposited from a suspension of *76YLa_milling* powder, had the highest tendency to crack upon drying.

### 4.2. Electrophoretic Deposition of One-Layer Compacts from Suspensions of HoLa Nanopowder

Preliminary experiments on EPD from a suspension of *HoLa* powder revealed that at an electric field strength of 20 V/cm, there was no deposition (mass growth on the electrode) from the *HoLa* suspension; therefore, subsequent experiments were carried out at a field strength of 40 V/cm. Figure 3 shows dependences of the current on the time of EPD from suspensions of *HoLa* nanopowder in the dispersion medium of isopropanol with acetylacetone (1 mg/m$^2$) without the addition of a binder (*HoLa_AcAc*) and with the addition of PVB (*HoLa_PVB*). From Figure 3, it can be seen that the magnitude of the current during the EPD changes insignificantly. The EPD current values for the *HoLa_AcAc* suspension ($I_{av}$ = 0.338 mA) are higher than for the *HoLa_PVB* suspension ($I_{av}$ = 0.184 mA).

Electrophoretic deposition of one-layer compacts *HoLa_1* and *HoLa _2* was carried out in a similar way from *HoLa_AcAc* and *HoLa_PVB* suspensions, respectively. From Table 2, it can be seen that the introduction of PVB into the suspension of *HoLa* nanopowder led, as in the case of suspensions of yttrium oxide nanopowders, to a decrease in the mass and thickness of the compact (sample *HoLa_2*).

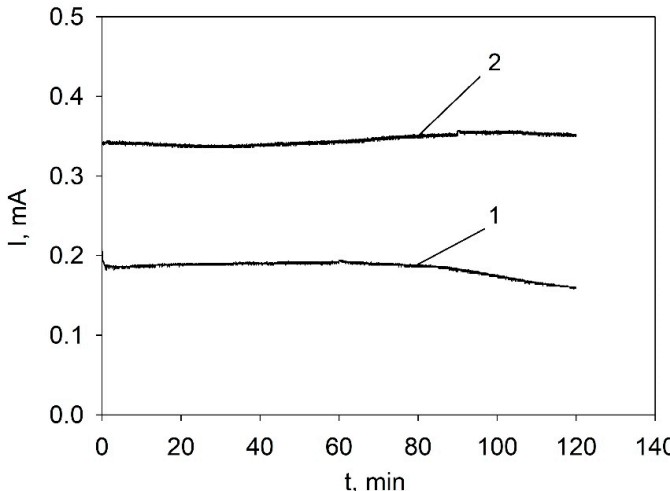

**Figure 3.** Dependences of the current on time during the EPD (electric field strength of 40 V/cm) from *HoLa_PVB* suspension with the addition of PVB (sample *HoLa_2*, curve 1) and *HoLa_AcAc* without the addition of a binder (sample *HoLa_1*, curve 2).

*4.3. Electrophoretic Deposition of Two-Layer Compacts*

Figure 4 shows the kinetics of the current during the electrophoretic deposition of the two-layer compacts *76YLa/HoLa_1* and *80YLa/HoLa_2*. It can be seen that during the EPD of the first *76YLa* layer of the *76YLa/HoLa_1* sample, the current slightly increases with increasing time from 0.145 to 0.162 mA (Figure 4a). During the EPD of the second *HoLa* layer, there is a fairly obvious tendency for the current to decrease by a factor of 1.8 from 0.294 to 0.166 mA, which is associated with an increase in the electrical resistance of the layer deposited on the cathode.

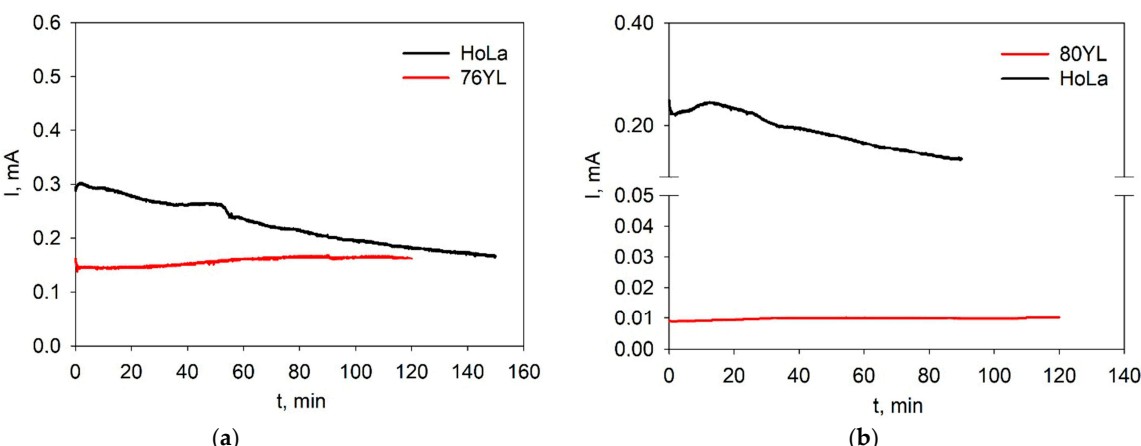

(**a**)                                                                                                  (**b**)

**Figure 4.** Dependences of the current on time during the EPD of two-layer samples: (**a**) *76YLa/HoLa_1* and (**b**) *80YLa/HoLa_2*.

In the case of deposition of the two-layer compact from suspensions with PVB during the EPD of the first *80YLa* layer of the *80YLa/HoLa_2* sample, the current remains almost constant (Figure 4b). At the stage of the EPD of the second *HoLa* layer, the current decreases by a factor of 2 from 0.250 to 0.135 mA.

The mass of the *80YLa/HoLa_2* sample was 456 mg with a thickness of 1.2 mm and a density of 46%, while the *76YLa/HoLa_1* sample with the same density of 46% was characterized by the largest mass, equal to 926 mg, and a thickness of 2.6 mm. Obviously, the increase in the mass and thickness of the *76YLa/HoLa_1* sample is due to the use of a suspension based on the *76Yla* powder without the addition of PVB during the deposition of the first layer (Table 3). As noted earlier in the case of the deposition of the one-layer

compacts, the introduction of the PVB into the suspensions of nanoparticles leads to a decrease in the mass and thickness of the samples.

Figure 5 shows a sample of the two-layer compact after deposition and drying. It can be seen that the sample is characterized by good uniformity of the deposited layers.

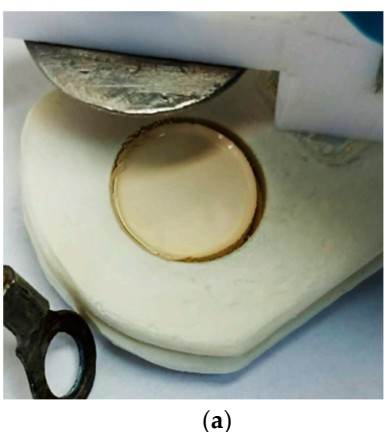 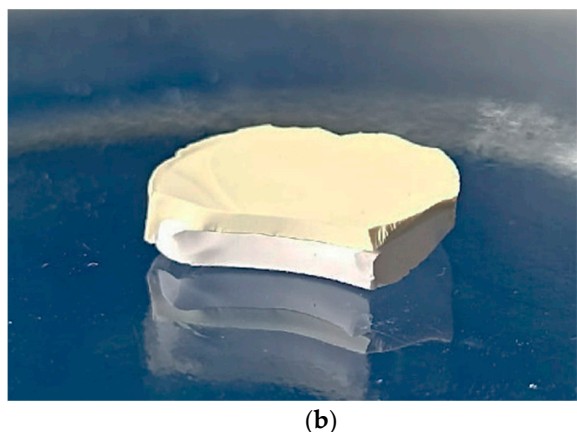

(**a**)                    (**b**)

**Figure 5.** Two-layer compact formed by EPD: (**a**) in the EPD cell and (**b**) after drying.

The obtained two-layer compacts were sintered in a vacuum at a temperature of 1750 °C for 10 h. The heating rate was 0.5 °C/min. The characteristic size of crystallites in the obtained ceramics was 15–20 μm. In samples where the *YLa* layer did not contain PVB (*76YLa/HoLa_1*), cracks appeared in the longitudinal plane after sintering. Samples that contained PVB in both layers remained uniform after the vacuum sintering. Figure 6a shows a micrograph of a cross-section of the *80YLa/HoLa_2* sample. The distribution of chemical elements over the depth of the sample is shown in Figure 6b. Since the images were obtained by collecting backscattered electrons, the darker region corresponds to materials with lower density—in this case, $(Y_{0.9}La_{0.1})_2O_3$, which has a density of 5.2 g/cm$^3$ (in comparison with $(Ho_{0.95}La_{0.05})_2O_3$—8.3 g/cm$^3$). The thickness of the layer with interdiffused ions is about 30 μm.

The resulting ceramics demonstrate complete sintering of the *YLa/HoLa* layers, low porosity of the *HoLa* layer, and a relatively high content of pores with characteristic sizes of a few microns at crystallite boundaries in the *YLa* layer (Figure 6a). We believe the difference in the porosity to be caused by differences in diffusion rates (primarily, surface diffusion) in the materials. It is known that such pores in optical ceramics can be eliminated by either significantly increasing the sintering time or using additional hot isostatic pressing (HIP) [15,30]. Two facts are worth additional discussion:

1. There are micron-sized pores located almost in the same plane near the *YLa/HoLa* interface. It is possible that the appearance of these pores is associated with air bubbles that appeared on the surface of the deposited *YLa* layer at the moment of transfer of the sample to the *HoLa* suspension and remained in the two-layer compact after drying. In this case, their appearance can be avoided by adjusting the process of the sample movement between the suspensions.

2. If the assumption that the pores are located at the boundary of the deposited layers is correct, then it follows from the ion distributions (Figure 6b) that during the sintering, predominant (almost one-sided) diffusion of holmium ions into the $Y_2O_3$ lattice occurs. Taking into account the proximity of the ionic radii of Ho$^{3+}$ and Y$^{3+}$ in the cubic lattice of the sesquioxides (1.015 Å and 1.019 Å, respectively), as well as the presence of La$^{3+}$ ions and the fact that during the sintering, diffusion proceeds primarily along crystallite boundaries, weak diffusion of yttrium ions into the *HoLa* layer is difficult to explain. It is possible that such a shift in the distribution of chemical elements is associated not with the diffusion processes during the sintering, but with the incorporation of *HoLa* nanoparticles into the *YLa* layer at the initial stage of the EPD of the second layer. Indeed, the *HoLa*

nanoparticles accelerated by an electric field in suspension can be driven deep into the *YLa* layer deposited at the first stage of the EPD.

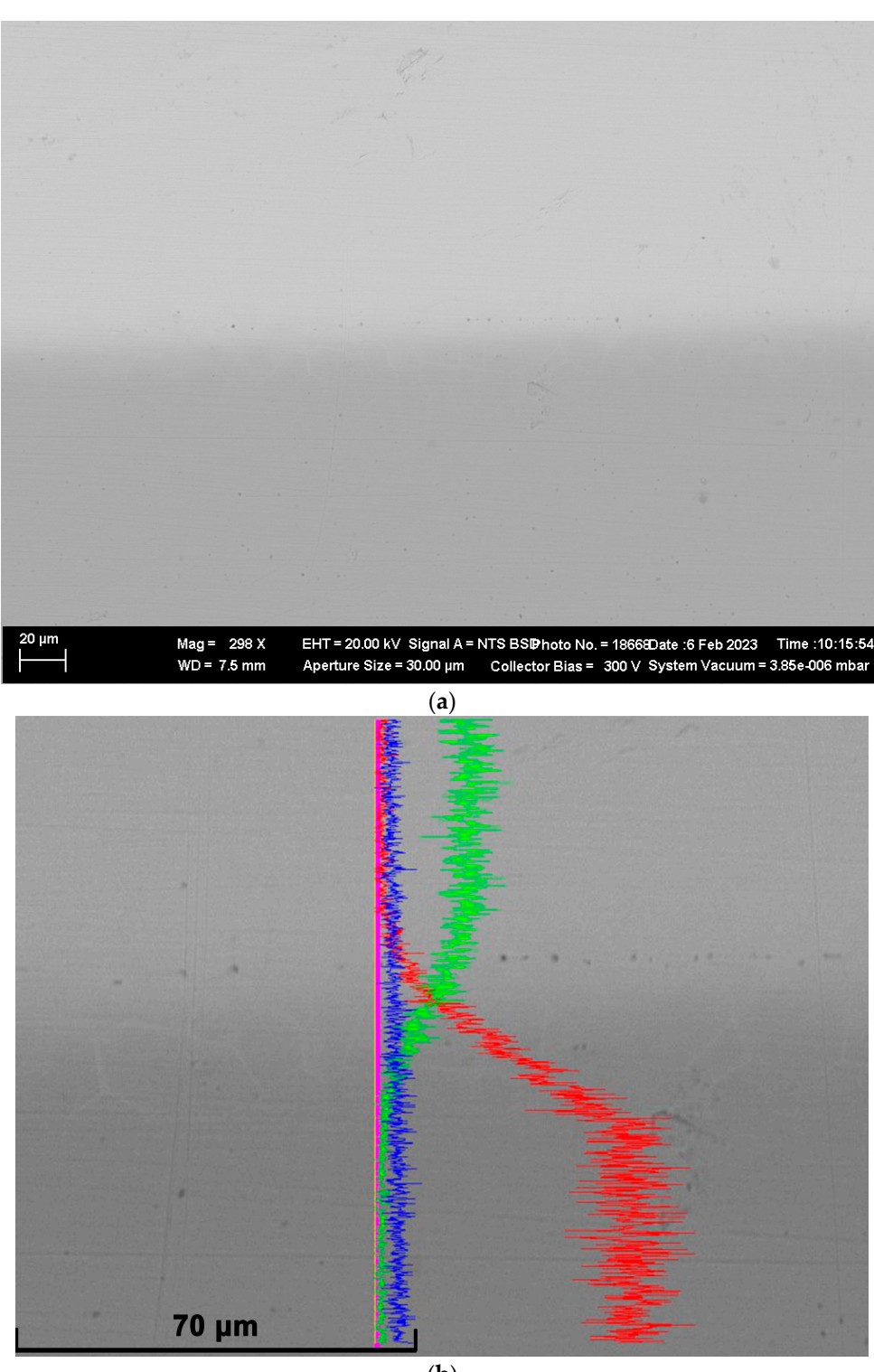

**(a)**

**(b)**

**Figure 6.** Micrograph of cross-section of the *80YLa/HoLa_2* sample (**a**) and the distribution of chemical elements (**b**): yttrium (red), holmium (green), and lanthanum (blue).

## 5. Conclusions

The use of acetylacetone and PVB made it possible to obtain stable suspensions of $(Y_{0.9}La_{0.1})_2O_3$ and $(Ho_{0.95}La_{0.05})_2O_3$ nanopowders, characterized by high $\zeta$–potential

(+29…+80 mV), which provides a chance to carry out a stable EPD process and make a bulk ceramic green body. One-layer compacts of the *YLa* and *HoLa* nanopowders with a density of 30–43% were formed from nanopowder suspensions by electrophoretic deposition. It was found that the introduction of PVB into the suspensions leads to a decrease in the mass and thickness of the compacts, but does not significantly affect their density. Two-layer *YLa/HoLa* compacts were made by layer-by-layer electrophoretic deposition. It was shown that with the help of the EPD method, it is possible to form two-layer compacts with a thickness of up to 2.6 mm and a density of up to 46%. Sintering such compacts in a vacuum at a temperature of 1750 °C leads to the formation of ceramics with a homogeneous boundary between the *YLa/HoLa* layers, where the thickness of the layer with interdiffused ions is about 30 μm. The ceramics show residual porosity in the *YLa* layer and at the interface between the deposited *YLa/HoLa* layers. It can be expected that these pores with a characteristic size of a few microns will be eliminated with an increase in sintering time or additional HIP treatment of the ceramics.

**Author Contributions:** Conceptualization, E.G.K. and M.G.I.; methodology, E.G.K. and M.G.I.; formal analysis, E.G.K. and M.G.I.; investigation, E.G.K., N.D.K. and D.K.K.; resources, E.G.K., M.G.I., N.D.K. and D.K.K.; data curation, E.G.K. and M.G.I.; writing—original draft preparation, E.G.K.; writing—review and editing, M.G.I.; visualization, E.G.K.; project administration, M.G.I.; funding acquisition, M.G.I., N.D.K. and D.K.K. All authors have read and agreed to the published version of the manuscript.

**Funding:** This study was funded by the Russian Science Foundation, research project No. 18-13-00355, https://rscf.ru/en/project/18-13-00355 (accessed on 19 July 2023). A part of the research performed by Kundikova N.D. was funded by the Ministry of Science and Higher Education (No. 122011200363-9). The scanning electron microscopy, performed in the Ural Center for Shared Use "Modern nanotechnology" at Ural Federal University (Reg.№ 2968) was funded by the Ministry of Science and Higher Education (Project № 075-15-2021-677).

**Institutional Review Board Statement:** Not applicable.

**Informed Consent Statement:** Not applicable.

**Data Availability Statement:** Data are contained within the article.

**Conflicts of Interest:** The authors declare no conflict of interest.

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
