# Peer review of "Electrophoretic Deposition of One- and Two-Layer Compacts of Holmium and Yttrium Oxide Nanopowders for Magneto-Optical Ceramics Fabrication"

_magnetochemistry, doi:10.3390/magnetochemistry9110227_

Round 1

Reviewer 1 Report

Comments and Suggestions for Authors

E.G. Kalinina et al reported the electrophoretic deposition of Ho and Y ceramics from oxide suspensions. The authors show proper experimental design and detailed discussions of results. The paper is well written and organized. Here are some minor suggestions:

1. Line 45, there should be reference(s) for the Nd:YAG/sapphire example.

2. Full names of the abbreviations should be properly mentioned: iPrOH, AcAc

3. Some experiments were not described precisely: Line 145, the power/frequency/speed of the ultrasonic probe? Line 196 and 204, how long is “several days”? Line 209, what is “pressure of no more than 10-2 Pa”? Please state the exact pressure.

4. What is the aim of doing the aging experiment? What is the aging condition?

5. Pores at the boundaries are hard to see in Figure 6a. Do the authors have magnified images?

6. A brief introduction of research background should be included in the abstract before mentioning what has been done in this work.

Author Response

Dear reviewer!

Thank you very much for your effort and, of course, the encouraging comments on our manuscript, but also for pointing out our shortcomings. Some of the faults have obviously been overlooked despite re-readings, other issues have escaped our attention. We have corrected the manuscript in full accordance with your comments. Changes in the text of the manuscript are marked in green.

E.G. Kalinina et al reported the electrophoretic deposition of Ho and Y ceramics from oxide suspensions. The authors show proper experimental design and detailed discussions of results. The paper is well written and organized. Here are some minor suggestions:

  1. Line 45, there should be reference(s) for the Nd:YAG/sapphire example.

The refences were added.

  1. Full names of the abbreviations should be properly mentioned: iPrOH, AcAc

Corrected.

  1. Some experiments were not described precisely: Line 145, the power/frequency/speed of the ultrasonic probe? Line 196 and 204, how long is “several days”? Line 209, what is “pressure of no more than 10-2 Pa”? Please state the exact pressure.

The information has been added to the manuscript.

  1. What is the aim of doing the aging experiment? What is the aging condition?

The technology of electrophoretic deposition is based on the preparation of stable suspensions of powders, which allow EPD to be carried out using previously prepared suspensions. From a practical point of view, an important task is to study the aging of suspensions in terms of changes in their properties and characteristics of the resulting compacts. During the study of aging, the suspension was stored in a dark place at room temperature for 14 days.

The information has been added to the manuscript.

  1. Pores at the boundaries are hard to see in Figure 6a. Do the authors have magnified images?

Unfortunately, due to technical and organizational reasons we can’t make the pictures with higher magnification, but the picture is good enough to see the pores quite clearly. We guess the resolution of the image was lost in the .pdf file, thus we are going to add the high-resolution picture to the supplemental materials.

  1. A brief introduction of research background should be included in the abstract before mentioning what has been done in this work.

The abstract has been corrected.

Reviewer 2 Report

Comments and Suggestions for Authors

Authors should write shorter sentences at the beginning of the abstract.

Still in the Abstract, remove the reticence between the zeta potential values and replace it with the range of values obtained (initial value - final value).

The Introduction is well written and with current references for the description of concepts and the state of the art of the research topic. I suggest including references to the statements in the first paragraph.

In the Materials and Methods section, only the description of materials and methods used should remain. The results obtained from the techniques should be in the next section (Results).

There is a explanation/discussion within the methodological section, which does not need to be there. Authors can start the results section by showing this data. For example, the suspension stability data in Table 1 (which is actually discussed in the Results section).

The characterizations of the suspensions were given in detail, as well as the temporal evolution of the directly measured current.

In any case, the authors must inform the number of samples being used in each experiment.

The authors should better explain why the sintering temperatures used.

What the authors describe as existing in Fig. 6a, is not seen in the picture. There is no way to guarantee what the authors describe to exist there. The same goes for the description of what happens in Fig. 6b. Authors should improve the description.

The work appears to be current, interesting to the area, but these details need to be reviewed.

Author Response

Dear reviewer!

Thank you very much for your effort and, of course, the encouraging comments on our manuscript, but also for pointing out our shortcomings. Some of the faults have obviously been overlooked despite re-readings, other issues have escaped our attention. We have corrected the manuscript in full accordance with your comments. Changes in the text of the manuscript are marked in green.

Authors should write shorter sentences at the beginning of the abstract.

The abstract has been corrected.

Still in the Abstract, remove the reticence between the zeta potential values and replace it with the range of values obtained (initial value - final value).

Corrected.

The Introduction is well written and with current references for the description of concepts and the state of the art of the research topic. I suggest including references to the statements in the first paragraph.

The references were added.

In the Materials and Methods section, only the description of materials and methods used should remain. The results obtained from the techniques should be in the next section (Results).

Corrected.

There is a explanation/discussion within the methodological section, which does not need to be there. Authors can start the results section by showing this data. For example, the suspension stability data in Table 1 (which is actually discussed in the Results section).

Corrected.

The characterizations of the suspensions were given in detail, as well as the temporal evolution of the directly measured current. In any case, the authors must inform the number of samples being used in each experiment.

Corrected.

The authors should better explain why the sintering temperatures used.

As our previous research demonstrated, the sintering conditions give a chance to make ytterbium doped (Y0.9La0.1)2O3 laser ceramics and (Ho0.95La0.05)2O3 magnetooptical ceramics where the lanthana was used as a sintering aid. The explanation has been added to the manuscript.

What the authors describe as existing in Fig. 6a, is not seen in the picture. There is no way to guarantee what the authors describe to exist there. The same goes for the description of what happens in Fig. 6b. Authors should improve the description.

Unfortunately, due to technical and organizational reasons we can’t make the pictures with higher magnification, but the picture is good enough to see the pores quite clearly. We guess the resolution of the image was lost in the .pdf file, thus we are going to add the high-resolution picture to the supplemental materials. Concerning the discussion of the picture, you are absolutely right, we can’t guarantee that our explanation is the ultimate truth. This is the reason why we use here such phrases as: “If the assumption… is correct”, “It is possible that…”

The work appears to be current, interesting to the area, but these details need to be reviewed.

Thank you for your comments and kind opinion!

Reviewer 3 Report

Comments and Suggestions for Authors

The manuscript describes the fabrication of one- and two-layers compacts of (Ho,La)2O3 and (Y,La)2O3 composites by electrophoretic deposition for magneto-optical application. The idea is interesting and the researchers carried out preliminary exploration. Though making a fully transparent composite material is a challenge, I hope the researchers could achieve good result in the future. Herein, I have several questions as follows for the researchers’ consideration.

(1) The two-layers composite materials contain low porosity in the HoLa layer but a relatively high content of pores on grain boundaries in the YLa layer. Could the researchers explain the inconsistent status?

(2) The purpose of this work should be emphasized by a comprehensive review of related literatures on transparent magneto-optical Ho2O3 or Ho-based ceramic materials.

(3) I hope the researchers can provide appearance photos of the products, though the composites may be not yet transparent up to now. It does not matter. I hope the researchers could achieve good result in the future via optimized sintering or post-HIP treatment.

(4) The format of references should conform to journal criterion.

Comments on the Quality of English Language

Minor editing of English language is required.

Author Response

Dear reviewer!

Thank you very much for your effort and, of course, the encouraging comments on our manuscript, but also for pointing out our shortcomings. Some of the faults have obviously been overlooked despite re-readings, other issues have escaped our attention. We have corrected the manuscript in full accordance with your comments. Changes in the text of the manuscript are marked in green.

(1) The two-layers composite materials contain low porosity in the HoLa layer but a relatively high content of pores on grain boundaries in the YLa layer. Could the researchers explain the inconsistent status?

We believe the difference in the porosity to be caused by difference in diffusion rates (primarily, surface diffusion) in the materials. The explanation was added to the text. Unfortunately, at this stage of the research we can’t prove it with precise measurements.

(2) The purpose of this work should be emphasized by a comprehensive review of related literatures on transparent magneto-optical Ho2O3 or Ho-based ceramic materials.

We corrected the Introduction part and added references to the latest publications on the material.

(3) I hope the researchers can provide appearance photos of the products, though the composites may be not yet transparent up to now. It does not matter. I hope the researchers could achieve good result in the future via optimized sintering or post-HIP treatment.

Due to organizational and technical reasons we can’t add the pictures of these samples to this paper. The ceramics were cut into pieces, sent to different Institutes and finally destroyed during a series of measurements, but we also hope to demonstrate good ceramics in the future via optimized sintering or post-HIP treatment. Thank you for the encouraging words!

(4) The format of references should conform to journal criterion.

We used Mendeley Reference Manager in this manuscript, so we hope the journal format on the references will be met.

Minor editing of English language is required.

The manuscript revised was corrected by a proofreading company. If, nevertheless, you can find any grammatical mistakes, please point them out. We will complain to the company.
